# Safety Assessment and Probiotic Potential Comparison of *Bifidobacterium longum* subsp. *infantis* BLI-02, *Lactobacillus plantarum* LPL28, *Lactobacillus acidophilus* TYCA06, and *Lactobacillus paracasei* ET-66

**DOI:** 10.3390/nu16010126

**Published:** 2023-12-29

**Authors:** Jui-Fen Chen, Ko-Chiang Hsia, Yi-Wei Kuo, Shu-Hui Chen, Yen-Yu Huang, Ching-Min Li, Yu-Chieh Hsu, Shin-Yu Tsai, Hsieh-Hsun Ho

**Affiliations:** 1Research Product Department, R&D Center, Glac Biotech Co., Ltd., Tainan City 744, Taiwan; juifen.chen@glac.com.tw (J.-F.C.); shawn.hsia@glac.com.tw (K.-C.H.); jim.huang@glac.com.tw (Y.-Y.H.); chingmin.li@glac.com.tw (C.-M.L.); yuchieh.hsu@glac.com.tw (Y.-C.H.); shin-yu.tsai@glac.com.tw (S.-Y.T.); 2Functional Investigation Department, R&D Center, Glac Biotech Co., Ltd., Tainan City 744, Taiwan; vic.kuo@glac.com.tw; 3Process Department, R&D Center, Glac Biotech Co., Ltd., Tainan City 744, Taiwan; water.chen@glac.com.tw

**Keywords:** *Bifidobacterium*, *Lactobacillus*, safety, antimicrobial

## Abstract

*Bifidobacterium longum* subsp. *infantis* BLI-02, *Lactobacillus paracasei* ET-66, *Lactobacillus plantarum* LPL28, and *Lactobacillus acidophilus* TYCA06, isolated from healthy breast milk, miso, and the healthy human gut, were assessed for safety in this study. BLI-02, LPL28, TYCA06, and ET-66 exhibited no antibiotic resistance and mutagenic activity in the Ames test at the highest dosage (5000 μg/plate). No genotoxicity was observed in micronucleus and chromosomal aberration assays in rodent spermatogonia at the maximum dosage of 10 g/kg body weight (BW). No acute and sub-chronic toxicity occurred in mice and rats at the maximum tested dosage of 10 g/kg BW and 1.5 g/kg BW, respectively. The lyophilized powder of these strains survived a low pH and high bile salt environment, adhering strongly to Caco-2 cells. Unique antimicrobial activities were noted in these strains, with BLI-02 demonstrating the best growth inhibition against *Vibrio parahaemolyticus*, LPL28 exhibiting the best growth inhibition against *Helicobacter pylori*, and ET-66 showing the best growth inhibition against *Aggregatibacter actinomycetemcomitans*. Based on the present study, the lyophilized powder of these four strains appears to be a safe probiotic supplement at tested dosages. It should be applicable for clinical or healthcare applications.

## 1. Introduction

Lactic acid bacteria have been recognized as bacteria safe for human consumption under the Generally Recognized as Safe (GRAS) classification [1]. The historical use of lactic acid bacteria in food underscored the perception that the majority of strains were symbiotic microorganisms without pathogenic potential [2]. As awareness of nutritional health grows, there has been a flourishing development in products related to lactic acid bacteria and functional studies. Currently, *Bifidobacterium* and *Lactobacillus* are the most extensively researched lactic acid bacteria strains. Nevertheless, the 2002 report from the FAO/WHO Joint Expert Consultation emphasized the need to establish systematic methods for evaluating probiotics in food to substantiate health claims [3] despite the widespread perception of lactic acid bacteria as safe. Therefore, lactic acid bacteria must undergo verification of resistance to antibiotics, toxicity, and genetic variability before asserting their safety.

*B. longum* is a Gram-positive anaerobic bacterium. As early as 1963, the literature documented the isolation of *B. longum* from human feces [4]. In 2002, Sakata et al., based on DNA genetic similarity comparisons, unified *B. longum*, *B. infantis*, and *B. suis* into a single species named “*B. longum*”. Three subspecies were subsequently established: “*longum*”, “*infantis*”, and “*suis*” [5]. *B. longum* is well known under the category of probiotics. This species is reported to have several potential beneficial effects on human health, particularly in relation to gut health and overall well-being. Some of the notable beneficial effects attributed to *B. longum* include improving digestive health, supporting the immune system, alleviating gastrointestinal disorders, promoting mood and mental health, managing allergy, ameliorating lactose intolerance, and counteracting oxidative stress [6,7,8,9]. *B. longum* subsp. *infantis* has been confirmed to possess the ability to metabolize the oligosaccharide in breast milk and has demonstrated anti-inflammatory properties, as well as the capacity to improve intestinal permeability [10]. *B. longum* subsp. *infantis* BLI-02 was a strain newly isolated from human breast milk in 2016 and exhibited multiple beneficial effects, such as ameliorating renal dysfunction, facilitating weight loss, reducing oxidative stress, and enhancing brain-derived neurotrophic factor [11,12,13,14,15]. However, these studies did not provide detailed discussions on the dosage limitations associated with the use of the *B. longum* strains employed.

*Lactobacillus plantarum* is commonly found in various fermented foods and is also used as a probiotic supplement. The initial *L. plantarum* with a complete sequence was strain WCFS1, which was isolated from human saliva [16]. *L. plantarum*, as a facultative fermentation species, stands out for its distinctive plant-related characteristics, given its capability to ferment and convert a variety of plant-derived raw materials [17]. This species has been associated with several potential beneficial effects on human health. Some of the notable beneficial effects attributed to *L. plantarum* include promoting digestive health, supporting the immune system, balancing gut microbiota, reducing inflammation, managing allergy, improving metabolic health, counteracting oxidative stress, promoting skin health, and promoting mood and mental health [18,19,20,21,22]. *L. plantarum* LPL28 was a young strain isolated from fermenting soybeans in 2011 and displayed a beneficial effect on oral health [23,24]. Despite the widespread application of *L. plantarum* in food [25], aquaculture [26], and agriculture [27], the safety of its dosage still needs to be verified in this strain.

*L. acidophilus* was initially identified in 1900 by Moro, isolated from infant feces, and officially designated as *L. acidophilus* [28]. Historically, due to challenges in distinguishing between types of *Lactobacillus*, probiotic products were unable to clearly specify the species or strain names. With the ongoing development and maturation of strain identification technology, it has been found that *L. acidophilus* comprises numerous strains [29]; however, discussions regarding their safety are relatively limited. *L. acidophilus* is commonly used as a probiotic in various dairy products, supplements, and fermented foods. It is well-studied and has been associated with several potential beneficial effects on human health. Some of the notable beneficial effects attributed to *L. acidophilus* include promoting digestive health, supporting the immune system, maintaining vaginal health, alleviating lactose intolerance, reducing antibiotic-associated diarrhea, managing cholesterol levels, managing allergies, ameliorating gastrointestinal disorders, and improving oral health [29,30,31,32,33]. *L. acidophilus* TYCA06 was a strain lately isolated from a healthy human gut in 2016 and displayed a beneficial effect on renal health [11,14]. Furthermore, *L. acidophilus* TYCA06 has been applied in the production of probiotics and collagen co-fermentation-derived postbiotics, effectively improving acne vulgaris [34]. This strain displayed a good potential for broader application, so the safety assessment for its dosage is essential.

*L. paracasei* was identified in 1989 via the results of DNA-DNA hybridization, revealing its genomic heterogeneity with *L. casei* subsp. *casei*. This led to its reclassification and renaming [35]. *L. paracasei* usually falls under the category of probiotics and is commonly found in various fermented dairy foods and probiotic products. This species has been associated with several potential beneficial effects on human health. Some of the notable beneficial effects attributed to *L. paracasei* include promoting digestive health, supporting the immune system, balancing gut microbiota, managing allergies, maintaining skin health, alleviating gastrointestinal disorders, promoting oral health, reducing oxidative stress, and improving bone health [36,37,38,39,40]. *L. paracasei* ET-66 was a strain currently isolated from human breast milk in 2016 and exhibited a beneficial effect on oral health [23,24,41,42]. Even though numerous reports in the past have demonstrated the absence of side effects from probiotics in the human body, the rising health consciousness and increased attention to dietary issues have prompted many countries to establish increasingly stringent food regulations. Therefore, there is a legitimate necessity to investigate the safety of dosage limitations.

The definition of probiotics is “live microorganisms that, when administered in adequate amounts, confer a health benefit on the host” [43]. In order to provide optimal effects, some strategies were developed to increase the likelihood of lactic acid bacteria reaching the lower intestinal tract alive. The process of lyophilization offers several benefits, such as the long-term preservation of bacterial cultures without the need for refrigeration and reduced weight and volume, which is valuable for storage and transportation. However, the process is complex and requires careful control of temperature, pressure, and time to ensure the viability and integrity of the lactic acid bacteria [44]. After consumption, the lactic acid bacteria encounter low pH and bile in the upper gastrointestinal tract. It is believed that probiotics exert beneficial effects by adhering and interacting with the gut environment, influencing the composition and activity of the gut microbiota [45]. Therefore, the lyophilized probiotic powder should exhibit tolerance to low pH (acidic conditions) and bile salts. In addition, the survival and activation of the strain should be evaluated to ensure the probiotic property was preserved by lyophilization.

In this study, the safety of these lactic acid bacteria strains was evaluated by the minimum inhibitory concentration (MIC) of antibiotics, and their lyophilized powder was assessed via in vitro and in vivo studies. First, the *Salmonella typhimurium* reverse mutation assay (Ames test) was carried out with or without metabolic activation in vitro. Next, genotoxicity was observed by micronucleus assay and chromosomal aberration test in rodents. Furthermore, acute and sub-chronic oral toxicity was recorded in ICR mice and SD rats. The probiotic property of these strains was analyzed via survival assay in a low pH followed by a high bile salt environment and adhesion assay in intestinal epithelial Caco-2 cells. Finally, different antimicrobial activities were characterized in these lactic acid bacteria strains.

## 2. Materials and Methods

### 2.1. Microorganisms

Lyophilized lactic acid bacteria powder was manufactured by Glac Biotech Co., Ltd. (Tainan, Taiwan). Serial dilutions of the powder were used to determine bacterial counts in peptone water on de Man, Rogosa, and Sharpe (MRS) agar for *Lactobacillus* and on MRS agar containing cysteine for *Bifidobacterium*. Plates were incubated at 37 °C for 48 h anaerobically. The lot number 51020200145 of *B. longum* subsp. *infantis* BLI-02 (BCRC 910812 = CGMCC 15212) contained 2.4 × 10^11^ colony forming units (CFU)/g. The lot number 51220190431 of *L. plantarum* LPL28 (BCRC 910536 = CGMCC 17954) contained 2.21 × 10^11^ CFU/g. The lot number 51220200022 of *L. acidophilus* TYCA06 (BCRC 910813 = CGMCC 15210) contained 2.07 × 10^11^ CFU/g. The lot number 51020210043 of *L. paracasei* ET-66 (BCRC 910753 = CGMCC 13514) contained 2.72 × 10^11^ CFU/g. The suggested daily consumption of the lyophilized powder was 0.1 g/20 kg body weight (BW) for infants and children, which is equivalent to 0.3 g/60 kg BW in adults.

### 2.2. Minimum Inhibitory Concentration (MIC) of Antibiotics

To determine the antibiotic susceptibility of the lactic acid bacteria strains, tests for minimum inhibitory concentration (MIC) were conducted by the Food Industry Research & Development Institute, Hsinchu City 30062, Taiwan using internationally recognized standardized methods [European Food Safety Authority (EFSA), 2012; International Organization for Standardization (ISO), 2010] for the following relevant antibiotics: gentamicin, vancomycin, streptomycin, tetracycline, erythromycin, clindamycin, chloramphenicol, ampicillin, and kanamycin [46,47].

### 2.3. Salmonella Typhimurium Reverse Mutation Assay (Ames Test)

The Ames test was conducted to evaluate the mutagenicity of the lyophilized powder with or without S9 activation in accordance with the preparation method of S9 in *Salmonella typhimurium*/Mammals Microsomal Enzyme Test, Organisation for Economic Co-operation and Development (OECD) Test Guideline (No. 471, 2020). Four strains of *Salmonella typhimurium*, TA97a, TA98, TA100, and TA1535, and one strain of *Escherichia coli* WP2urvA were used in this assay (Molecular Toxicology, Boone, NC, USA). Five dosages of lyophilized powder at 0.3125, 0.625, 1.25, 2.5, and 5 mg/plate were prepared for analysis. The sterile water was used as the negative control. The five positive control groups comprised 50 μg/plate of dexon (sodium p-dimethylamino-benzenediazo sulfonate, Chem Service, West Chester, PA, USA) and 1 μg/plate of methyl methanesulfonate (Fu Chen Chemical Reagents, Tianjin, China) for the −S9 comparisons. Ten (10) μg/plate of 2-aminofluorene (Sigma–Aldrich, St. Louis, MO, USA), 50 μg/plate of 1,8-dihydroxyanthraquinone (Sigma–Aldrich, St. Louis, MO, USA), and 200 μg/plate of cyclophosphamide (TCI, Tokyo, Japan) were comprised for the +S9 comparisons. Incubation was performed at 37 ± 1 °C for 48 h, and the number of colonies was counted in triplicate.

### 2.4. Acute Oral Toxicity Test

For each strain, twenty Specific Pathogen Free (SPF) ICR mice (18–22 g, Charles River Laboratories, Beijing, China), 10 males and 10 females, were observed and selected after adapting to the feeding condition. The concentration of lyophilized powder was 0.25 g/mL in sterile water, and the gavage dose was 10 g/kg BW with 40 mL/kg BW gavage volume. Before the experiment, the mice were fasted for 4 h, and the test substance was given once by gavage. The total dosage was equivalent to more than 2 × 10^12^ CFU/kg BW. The assay was performed according to the OECD Test Guideline for acute oral toxicity tests (No. 423, 2002). The physical status, weight change, and signs of poisoning and death were recorded for 14 days. The approval code of experimental animal ethics was JN.No20200710S0801030(146).

### 2.5. In Vivo Mouse Spermatogonial Chromosomal Aberration Test

To assess the genotoxicity of the lyophilized powder, the spermatogonial chromosomal aberration assay was conducted following the OECD Test Guideline (No. 483, 2016). Each group consisted of 5 SPF ICR male mice (25–35 g, Charles River Laboratories, Beijing, China) were randomly assigned. The negative control group received sterile water, while the positive control group received a dose of 0.04 g/kg BW cyclophosphamide. The lyophilized powder of each strain was administered via oral gavage at low, medium, and high dosages of 2.5, 5, and 10 g/kg BW, respectively. After 24 h, the animals were euthanized by cervical dislocation, and testicle samples were obtained. Intraperitoneal injection of colchicine (5 mg/kg BW, 10 mL/kg BW) was performed four hours prior to sacrifice. Both testes were extracted, fat was removed, and they were washed in a buffer solution. The tissue samples were fixed, centrifuged, and stained with Giemsa staining. The number of chromosomal aberrations was recorded in 500 metaphase cells per animal, and the types of chromosomal aberrations, along with their respective numbers and frequencies, were listed for each group. The approval code of experimental animal ethics was JN.No20200710S0801030(146).

### 2.6. In Vivo Mammalian Erythrocyte Micronucleus Test

The study followed the guidelines specified by the OECD Test Guideline (No. 474, 2016). Each group consisted of 5 male and 5 female SPF ICR mice (25–35 g, Charles River Laboratories, Beijing, China) randomly allocated for the experiment. The negative control group received sterile water, while the positive control group received a dose of 0.04 g/kg BW cyclophosphamide. The lyophilized powder of each strain was administered via oral gavage at low, medium, and high dosages of 2.5, 5, and 10 g/kg BW, respectively. The second administration was given 24 h after the first one, and the mice were euthanized 6 h after the second administration. The bone marrow samples were collected, fixed, and examined under a microscope. The ratio of polychromatic erythrocytes (PCE) to normochromatic erythrocytes (NCE) (P/N) was observed in 200 erythrocytes per animal. The micronuclei, which are small additional nuclei formed during cell division, were counted in 2000 PCE per animal. The approval code of experimental animal ethics was JN.No20200710S0801030(146).

### 2.7. Repeated Dosage for 90-Day Oral Toxicity in Rats

The 90-day oral toxicity assay in the SPF SD rats was conducted in compliance with the OECD Test Guideline (No. 408, 2018). SD rats were subjected to constant conditions of humidity (40~70%) and temperature (20~26 °C). Following quarantine and acclimation for 1 week, each group consisted of 10 males and 10 females was randomly assigned. Sterile water was used as solvent control, and three different dose groups were set up: the low-(0.25 g/kg BW), medium-(0.5 g/kg BW), and high-dosage (1.5 g/kg BW) groups. During the experiment, samples were administered by oral gavage, and the animals were free to eat and drink. The general physical signs, behavior, toxic manifestations, and mortality of the animals were observed and recorded daily. The body weight and food consumption were measured weekly. At the end of the experiment, rats were sacrificed under ether anesthesia, and the blood was collected via the abdominal aorta. The blood sample was analyzed for hemoglobin (HB), red blood cells (RBC), hematocrit (HCT), white blood cells (WBC), platelet count (PLT), lymphocytes (LYMPH), neutrophil, acidophil, and basophil by a BC-5000 hematology analyzer (Mindray, Shenzhen, China). Activated partial thromboplastin time (APTT) and prothrombin time (PT) were analyzed by an SF-8050 automatic coagulation analyzer (Succeeder, Beijing, China). The plasma clinical chemistry parameters were evaluated for alanine aminotransferase (ALT), aspartate aminotransferase (AST), Alkaline phosphatase (ALP), γ-glutamyl transferase (γ-GT), urea, creatinine (CRE), glucose (GLU), total protein (TP), albumin (ALB), total cholesterol (TC), triacylglycerol (TG), chloride (Cl), potassium (K), and sodium (Na) by an AC9800 automatic electrolyte analyzer (Audicom, Jiangsu, China). The heart, thymus, adrenal glands, liver, kidneys, spleen, testis, ovaries, epididymis, uterus, and brain were collected for pathological checks. The relative organ weights were calculated by the following formula: relative organ weight = absolute organ weight (g)/body weight (g) × 100%. The approval code of experimental animal ethics was JN.No20200710S0801030(146).

### 2.8. Acid and High Bile Salt Tolerance of Lactic Acid Bacteria

The experiment involved the use of an acidified MRS medium (Difco, Detroit, MI, USA) to mimic gastric acid conditions. To prepare the medium, 27.5 g of MRS powder was dissolved in 450 mL of reverse osmosis (RO) water. The pH of the solution was then adjusted to 3.5 by adding 1N HCl solution. To simulate intestinal juice, the MRS medium containing 0.3% bile salt was used. This was prepared by dissolving 1.5 g of bile salt (Sigma, St. Louis, MO, USA) in 35 mL of autoclaved MRS medium (standard, non-acidified MRS). Probiotic powder was dissolved in 5 mL of the MRS medium, and colonies were counted on the plate as the reference CFU at 0 h. The same concentration of probiotic powder was dissolved in seven tubes of acidified MRS medium and incubated at 37 °C for 1–3 h. At each time point, one tube was taken and centrifuged at 4000 rpm for 10 min. The supernatant was discarded, and the pellet was washed twice with 5 mL of RO water. The pellet was then resuspended in 5 mL of RO water, and colonies were counted on a plate at 1, 2, and 3 h, respectively. At the third hour, four other tubes were centrifuged at 4000 rpm for 10 min, and the supernatant was discarded. These four tubes were incubated at 37 °C for an additional 1–4 h, with one tube taken at each time point. The culture was centrifuged at 4000 rpm for 10 min, and the supernatant was discarded. Colonies from each tube were counted on a plate at 4, 5, 6, and 7 h, respectively.

### 2.9. In Vitro Adherence Assay of Viable LACTIC Acid Bacteria to Intestinal Caco-2 Cells

The Caco-2 cells (BCRC 60182) were cultured at 37 °C with 5% CO_2_ in Dulbecco’s Modified Eagle Medium (DMEM, Danaher, Washington, DC, USA) supplemented with 10% fetal bovine serum (FBS, Invitrogen, Waltham, MA, USA). Cells were transferred to six-well plates (BD Falcon; Becton Dickinson) at a density of 1.5 × 10^5^ cells/mL and incubated until a complete monolayer formed. The medium was replaced every 48 h. Prior to co-culturing with bacteria, fresh DMEM was added to each well and incubated for 1 h. Overnight bacterial cultures were diluted with DMEM to a concentration of 5 × 10^8^ cells/mL, and viable bacteria were directly added to the Caco-2 cell culture. After incubating for 2 h at 37 °C, wells were rinsed twice with 1 × PBS to remove nonadherent bacteria. Cells were fixed with cold methanol for 5–10 min at room temperature, followed by Gram staining using the manufacturer’s instructions (Merck, Darmstadt, Germany). Adherent bacterial cells were counted in 20 random microscopic fields using a CX41 microscope (Olympus America, Inc., Bartlett, TN, USA).

### 2.10. In Vitro Bacteriostatic Activity Assay of Lactic Acid Bacteria

#### 2.10.1. The Modified Agar Overlay Method

The probiotic strains were streaked over MRS agar plates with a cotton swab and incubated at 37 °C for 48 h under either semi-anaerobic or anaerobic conditions to produce a 2-cm-wide probiotic growth zone. Subsequently, probiotic colonies were overlaid with 45 °C tryptone soy agar (TSA, BD Bioscience, Franklin Lakes, NJ, USA) supplemented with 2.5% NaCl (Sigma, St. Louis, MO, USA) for the culture of *Vibrio parahaemolyticus* or 45 °C brain heart infusion (BHI, BD Bioscience, Franklin Lakes, NJ, USA) for the culture of *Aggregatibacter actinomycetemcomitans*. Once the agar solidified, the prepared pathogenic culture was inoculated over the agar surface. The plate was then incubated at 37 °C for 48 h. After sufficient growth, the zones of inhibition were measured.

#### 2.10.2. The Liquid Culture Assays

The probiotic strains were cultured in MRS broth at 37 °C until the third generation, and the density of probiotics was adjusted to 1 × 10^9^ CFU/mL for co-culture with pathogens in the pathogens’ growth broth. The tubes containing the co-cultures were incubated at 37 °C for 48 h, and the survival rates of the pathogens were determined. After adequate growth, the inhibition scores of the pathogens were measured using plate counting.

### 2.11. Statistical Analysis

SPSS20.0 software was used to test the homogeneity of variance of the original data of each experiment, and the data that met the requirement of homogeneity of variance were statistically processed by the pairwise comparison method of the means between multiple experimental groups and a control group in the one-way analysis of variance (ANOVA) method. Perform appropriate variable transformation on the data with non-normal distribution or uneven variance. After meeting the requirements of normality or homogeneity of variance, use the converted data for statistical processing and use the rank-sum test for non-normally distributed data.

## 3. Results

### 3.1. The Antibiotic Resistance of B. longum *subsp*. infantis BLI-02, L. plantarum LPL28, L. acidophilus TYCA06, and L. paracasei ET-66

*B. longum* subsp. *infantis* BLI-02, *L. plantarum* LPL28, and *L. acidophilus* TYCA06 were susceptible to all tested antibiotics at a concentration below or equal to the EFSA cut-off values in the corresponding species (Table 1). *L. paracasei* ET-66 was susceptible to most of the tested antibiotics except for kanamycin and chloramphenicol. The MIC values were 128 mg/mL for kanamycin and 8 mg/mL for chloramphenicol in *L. paracasei* ET-66. The EFSA cut-off values were 64 mg/mL for kanamycin and 4 mg/mL for chloramphenicol in the species of *L. paracasei*. Additional genomic mining, including analysis using the Virulence Factor Database (VFDB setB) and the Comprehensive Antibiotic Resistance Database (CARD), was performed in *L. paracasei* ET-66 (Appendix A).

### 3.2. B. longum *subsp*. infantis BLI-02, L. plantarum LPL28, L. acidophilus TYCA06, and L. paracasei ET-66 Displayed No Mutagenic Activity In Vivo

The strains on all plates grew well, and no contamination colonies were observed. The results of the four lactic acid bacterial trains are shown in Figure 1. The results of the negative control group showed that the number of spontaneously reverted colonies of the four *S. typhimurium* strains was consistent with the reference value in the OECD standard. The identification of the colony indicated that the strain status was normal, and the number of reverted colonies in the positive control group was significantly higher than that of the spontaneous mutation, indicating that the positive mutagen was effective (*** *p* < 0.001). Comparing the results of different doses of lyophilized probiotic powder groups with the negative control group, the results showed that the number of reverted colonies of all doses of the sample group was not more than two times the number of reverted colonies of the negative control group both in the presence and absence of S9 mixture.

### 3.3. B. longum *subsp*. infantis BLI-02, L. plantarum LPL28, L. acidophilus TYCA06, and L. paracasei ET-66 Displayed No Acute Oral Toxicity In Vitro

During the test period, the animals in each group had normal diets and activities, grew well, did not see any signs of poisoning, no death, and no obvious abnormal changes in the weight of the mice (Table 2). The results showed that the acute oral toxicity LD_50_ of the lyophilized probiotic powder to male and female ICR mice was greater than 10 g/kg BW.

### 3.4. B. longum *subsp*. infantis BLI-02, L. plantarum LPL28, L. acidophilus TYCA06, and L. paracasei ET-66 Displayed neither Cytotoxicity to Bone Marrow nor Chromosomal Aberration Effect on Mouse Spermatogonia

The percentage of polychromatic erythrocytes (PCE) and total erythrocytes in each dose group of the lyophilized probiotic powder was not less than 20% of that in the negative control group, indicating that the sample had no obvious cytotoxicity at each dose (Figure 2, right *y*-axis). The rate of micronucleated reticulocytes (MN-RET) in the positive control group was higher than that of the negative control group (*** *p* < 0.001), indicating that the tested animals were sensitive, and the test was reliable (Figure 2, left *y*-axis). Compared with the negative control group, there was no significant difference in the rate of MN-RET in each dose group of the lyophilized probiotic powder (*p* > 0.05), suggesting that the sample has no micronucleus effect on mouse bone marrow cells within the tested dose range.

The chromosomal aberration rate of spermatogonial cells in the positive control group was significantly higher than that in the negative control group (** *p* < 0.01), indicating that the tested animals were sensitive, and the test was reliable (Table 3). Compared to the negative control group, there was no significant difference in the chromosomal aberration rate in each dose group of the lyophilized probiotic powder (*p* > 0.05), suggesting that the sample has no chromosomal aberration effect on mouse spermatogonia within the dose range.

### 3.5. B. longum *subsp*. infantis BLI-02, L. plantarum LPL28, L. acidophilus TYCA06, and L. paracasei ET-66 Displayed No Chronic Oral Toxicity

Rats were orally administered different doses of the lyophilized probiotic powder, including *B. longum* subsp. *infantis* BLI-02, *L. plantarum* LPL28, *L. acidophilus* TYCA06, or *L. paracasei* ET-66, for a duration of 90 days. No obvious abnormalities were found in the appearance, behavior, and feces of the animals in each dose group and the negative control group. Figure 3 illustrates the recorded animal weights over the 90-day period. In Figure 3A for BLI-02, Figure 3B for LPL28, Figure 3C for TYCA06, and Figure 3D for ET-66, there were no significant differences in the weekly average body weight, weekly weight gain, and total weight gain of both male and female animals in each dose group of the sample compared with the negative control group (*p* > 0.05). There was no statistical difference in the weekly food intake and total food intake of the animals (*p* > 0.05).

The wet weight of the heart, thymus, adrenal gland, liver, spleen, testis, ovary, epididymis, uterus, and brain was measured. The organ-to-body weight ratio of male and female rats in each dose group is displayed in Figure 4. There was no statistical difference between the lyophilized probiotic powder group and the negative control group (*p* > 0.05). Based on the above, different doses of 1.5 g/kg BW, 0.5 g/kg BW, and 0.25 g/kg BW were continuously orally administered to SD rats for 90 days, and no adverse effect of the lyophilized probiotic powder was observed on the appearance and behavior of the animal.

### 3.6. B. longum *subsp*. infantis BLI-02, L. plantarum LPL28, L. acidophilus TYCA06, and L. paracasei ET-66 Displayed No Effects on Hematological and Blood Biochemistry Parameters

After giving different doses of the lyophilized probiotic powder orally for 90 days, the blood sample of SD rats was collected and analyzed. The results of blood hemoglobin concentration, red blood cell count, hematocrit, platelet count, and white blood cell count are listed in Table 4. The composition of white blood cells was observed by determining the percentages of lymphocytes, neutrophils, eosinophils, and basophils in the total number of white blood cells. To assess the coagulation of blood, prothrombin time and activated partial thromboplastin time were investigated. All results showed no statistical difference in each dose group compared with the negative control group (*p* > 0.05).

The result of serum biochemistry parameters is listed in Table 5. The serum alanine aminotransferase (ALT), aspartate aminotransferase (AST), alkaline phosphatase (ALP), γ-glutamyl transferase (γ-GT), urea (Urea), creatinine (Cr), blood glucose (Glu), total protein (TP), albumin (Alb), total cholesterol (TC), triglyceride (TG), chloride (Cl), potassium (K), and sodium (Na) was determined in each dosage group. Compared with the negative control group, no statistical difference was found in any group (*p* > 0.05).

### 3.7. B. longum *subsp*. infantis BLI-02, L. plantarum LPL28, L. acidophilus TYCA06, and L. paracasei ET-66 Displayed Probiotic Potential against Human Pathogens

A 7 h acid followed by bile salt challenge was carried out in vitro to estimate whether the oral supplementation of the lyophilized probiotic powder was able to travel through the stomach and sufficiently reach the intestine (Figure 5A). Among these four strains, *L. plantarum* LPL28 displayed the strongest tolerance to the challenge. The viable LPL28 bacterial cells decreased from 5.69 × 10^9^ CFU/mL to 2.04 × 10^9^ CFU/mL after 3 h in pH 3.5 and further decreased to 4.03 × 10^8^ CFU/mL after following 4 h in 0.3% bile salt. *B. longum* subsp. *infantis* BLI-02 displayed weaker tolerance to the challenge. The viable BLI-02 bacterial cells decreased from 2.56 × 10^9^ CFU/mL to 5.66 × 10^8^ CFU/mL after 3 h in pH 3.5 and further decreased to 3.03 × 10^7^ CFU/mL after following 4 h in 0.3% bile salt.

The co-culture with the human intestinal epithelial Caco-2 cell line was performed to evaluate whether the lyophilized probiotic powder was able to remain inside the intestinal tract (Figure 5B). Among these four strains, *L. paracasei* ET-66 displayed the best adhesion with 313.3 cells/field, and *L. acidophilus* TYCA06 had lower adhesion with 182.3 cell/field. Moreover, the bacteriostatic activity tests indicated these four lactic acid bacteria inhibited the growth of human pathogens (Table 6). *B. longum* subsp. *infantis* BLI-02 showed the best growth inhibition against *Vibrio parahaemolyticus*, which usually causes gastrointestinal illness in humans. *L. plantarum* LPL28 showed the best growth inhibition against *Helicobacter pylori*, which is usually associated with stomach illness. *L. paracasei* ET-66 showed the best growth inhibition against *Aggregatibacter actinomycetemcomitans*, which is suspected to be involved in chronic periodontitis. Taken together, the lyophilized powder displayed probiotic characteristics, which could pass through the gastrointestinal tract, surviving in lowering the intestinal tract, and exerting beneficial effects on human health.

## 4. Discussion

The misuse of antibiotics has made antibiotic resistance a serious issue for global public health, and resistance to many classes of antibiotics was reported in lactic acid bacteria [48,49]. The mechanisms of antimicrobial resistance in bacteria are widely diverse, ranging from unknown to well-studied [50]. To ensure the safe use of microorganisms as feed additives, the FEEDAP Panel has proposed microbiological cut-off values for distinguishing resistant from susceptible strains [46]. Among our strains, the MIC result fulfilled the basic requirement of the European Food Safety Authority (EFSA) in three strains: *B. longum* subsp. *infantis* BLI-02, *L. plantarum* LPL28, and *L. acidophilus* TYCA06. The MIC values were higher for kanamycin and chloramphenicol in *L. paracasei* ET-66. In a study involving 121 *L. paracasei* strains, it was demonstrated that MIC values commonly ranged from 32 to 128 mg/mL for kanamycin and from 8 to 16 mg/mL for chloramphenicol in this species [51]. Therefore, the MIC value of 128 mg/mL for kanamycin and 8 mg/mL for chloramphenicol in *L. paracasei* ET-66 may be attributed to intrinsic resistance. Our VFDB and CARD analyses showed good agreement, revealing a limited risk of acquired resistance in *L. paracasei* ET-66.

Using the acute oral toxicity test, it is confirmed that the oral toxicity LD_50_ of the lyophilized probiotic powder to female and male ICR mice is greater than 10 g/kg BW, which belongs to the actual non-toxic level [52]. Multiple tests are normally required to assess the genotoxicity of the material because no single assay can comprehensively detect different types of DNA damage [53,54]. In this study, three tests were conducted to ensure the safety of the lyophilized probiotic powder: the Ames test for reverse mutation in bacterial cells, the mammalian erythrocyte micronucleus test, and the mouse spermatogonial chromosomal aberration test. All the results from these genetic toxicity tests were negative, indicating that the lyophilized probiotic powder had no mutagenic effects within the tested dose range [55]. The health benefits of probiotics become noticeable when administered in adequate amounts, which is why long-term supplementation with probiotics is usually recommended [56]. In this study, the 90-day repeated dose study was performed to assess the safety of long-term consumption of our lyophilized probiotic powder. Throughout the 90-day period, SD rats exhibited no abnormal behavior and remained in good health when administered daily doses of 0.25 g/kg BW, 0.5 g/kg BW, and 1.5 g/kg BW. Animal weight, weight gain, and food utilization were similar in all groups. No adverse effects of the lyophilized probiotic powder on animal blood and blood biochemical parameters were observed. Eye examinations and gross anatomical examinations showed no obvious abnormalities in animals in each dose group. The lyophilized probiotic powder had no adverse effect on the wet weight and viscera-to-body ratio of the tested organs. Histopathological examination showed no histopathological changes related to the lyophilized probiotic powder in the tested organs. Taken together, no long-term toxic effect of *B. longum* subsp. *infantis* BLI-02, *L. plantarum* LPL28, *L. acidophilus* TYCA06, and *L. paracasei* ET-66 was found in this study [57].

To assess the probiotic potential of our strains, we compared their performance in the acid-bile salt challenge and adhesion assay against two well-established probiotic strains, *L. rhamnosus* GG (LGG) and *B. animalis* BB-12 (BB-12), which have a long history of use [58,59]. Our strains exhibited good acid and bile tolerance, whereas the viability of LGG decreased significantly during the transition from the acidic to the bile salt environment. Our strains exhibited good adhesion to Caco-2 cells, whereas the adhesion of BB-12 was a little above 100 cell/field. Nevertheless, this preliminary comparison warrants further investigation, as the lyophilized powder of LGG and BB-12 was produced by another company [60,61]. The adhesion condition and the lyophilization are two complex processes that require tailoring to specific strains [62,63]. Our study aimed to assess whether our lyophilized probiotic powder meets market criteria, with the understanding that the intricacies of the lyophilization process and adhesion conditions are outside the scope of this study.

In the bacteriostatic activity assay, our lyophilized probiotic powder inhibited the growth of *Vibrio parahaemolyticus*, *Helicobacter pylori*, and *Aggregatibacter actinomycetemcomitans*. Microorganisms can exhibit both attraction and antagonism toward each other, depending on various factors such as their ecological niche, competition for resources, and the specific microorganisms involved [64]. The antagonistic interactions among microorganisms are considered a promising alternative approach for treating bacterial infections and are being explored for applications in aquaculture or disease prevention in humans [65,66,67]. Numerous studies are underway to investigate the mechanisms behind antagonistic interactions, with a growing emphasis on characterizing bacteriocins. Bacteriocins are a type of antimicrobial protein or peptide produced by certain bacteria, and some probiotic bacteria are known to produce bacteriocins [68]. Future studies on characterizing and purifying the bacteriocins from our strains are recommended and would expand their applications beyond viable probiotics.

## 5. Conclusions

*B. longum* subsp. *infantis* BLI-02, *L. plantarum* LPL28, *L. acidophilus* TYCA06, and *L. paracasei* ET-66 exhibited a very low risk of antibiotic resistance and mutagenicity. Moreover, no signs of physical or genetic toxicity were observed in rodents. Consequently, the lyophilized powder containing these four strains is considered safe for use and has potential health benefits for humans.

## Figures and Tables

**Figure 1 nutrients-16-00126-f001:**
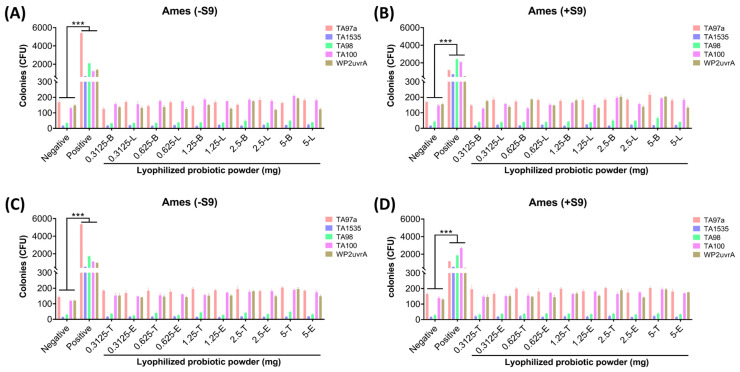
Mutagenic activity test (Ames test) indicated no mutagenic response was observed in *Salmonella typhimurium* strains TA97a, TA1535, TA98, TA100, and *Escherichia coli* WP2urvA treated with *B. longum* subsp. *infantis* BLI-02 (-B), *L. plantarum* LPL28 (-L), *L. acidophilus* TYCA06 (-T), and *L. paracasei* ET-66 (-E) at dosages of 0.3125, 0.625, 1.25, 2.5, and 5 mg/plate. The mutagenicity of *B. longum* subsp. *infantis* BLI-02 and *L. plantarum* LPL28 powder was investigated (**A**) without S9 activation and (**B**) with S9 activation. The mutagenicity of *L. acidophilus* TYCA06 and *L. paracasei* ET-66 powder was investigated (**C**) without S9 activation and (**D**) with S9 activation. Data are presented as mean ± SD of triplicate results. Sterile water was used in the negative control, and five mutagenic chemicals were used in the positive control. In groups without S9 activation, dexon was used as the mutagen for TA97a and TA98 (50 μg/plate), methyl methanesulfonate was used for TA100 and WP2urvA (1 μg/plate), and sodium azide was used for TA1535 (1.5 μg/plate). In groups with S9 activation, 2-aminofluorene was used as the mutagen for TA97a, TA98, TA100, and WP2urvA (20 μg/plate), and 2-aminoanthracene was used for TA1535 (2 μg/plate). *** *p* < 0.001 represents a significant difference compared with the negative control group. Abbreviation: CFU, colony-forming units.

**Figure 2 nutrients-16-00126-f002:**
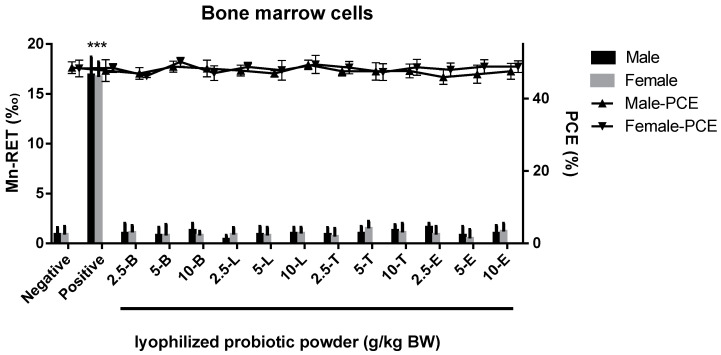
The micronucleus test showed that *B. longum* subsp. *infantis* BLI-02 (-B), *L. plantarum* LPL28 (-L), *L. acidophilus* TYCA06 (-T), and *L. paracasei* ET-66 (-E) were not cytotoxic to bone marrow following oral exposure in male and female mice at dosages of 2.5, 5, 10 g/kg BW. Numbers of micronucleated reticulocytes (MN-RET) per 1000 red blood cells (RBC) were indicated at left *y*-axis, and numbers of polychromatic erythrocytes (PCE) per 200 RBC were indicated at right *y*-axis. Sterile water was used as the negative control, and cyclophosphamide (0.04 g/kg BW) was used as the positive control. *** *p* < 0.001 represents a significant difference compared with the negative control group.

**Figure 3 nutrients-16-00126-f003:**
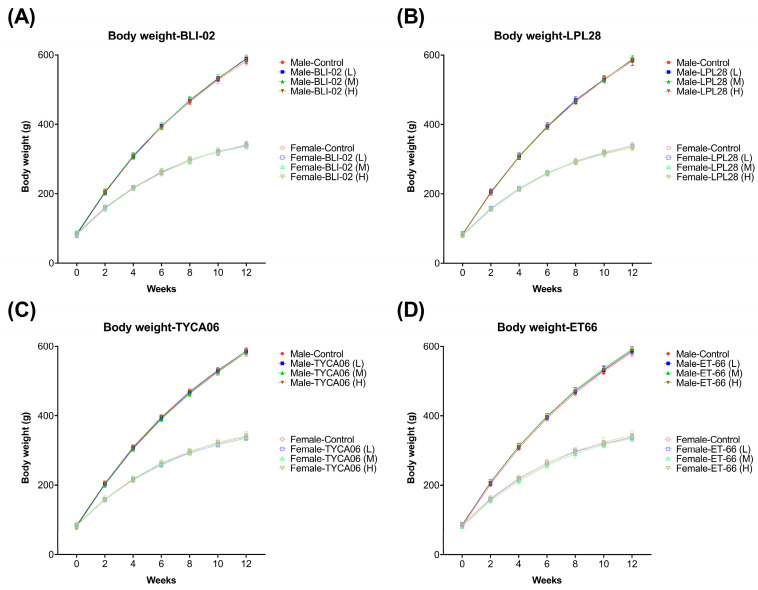
Probiotic powder administration had no statistically significant effect on the body weight during the 90-day oral toxicity test in (**A**) *B. longum* subsp. *infantis* BLI-02, (**B**) *L. plantarum* LPL28, (**C**) *L. acidophilus* TYCA06, and (**D**) *L. paracasei* ET-66 in both male (*n* = 10) and female (*n* = 10) animals. Via oral gavage, animals were administered lyophilized probiotic powder at dosages of 250 (Low, L), 500 (Medium, M), and 1500 (High, H) mg/kg BW. Animals were weighed, and body weights were recorded every two weeks for 90 days.

**Figure 4 nutrients-16-00126-f004:**
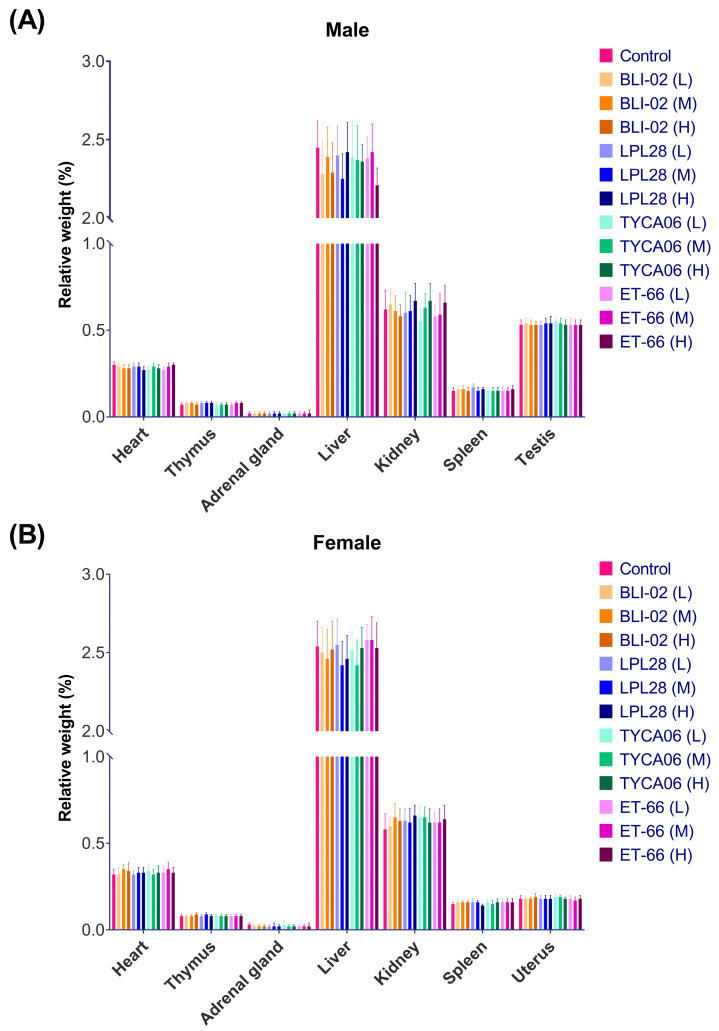
*B. longum* subsp. *infantis* BLI-02, *L. plantarum* LPL28, *L. acidophilus* TYCA06, and *L. paracasei* ET-66 powder administration had no statistically significant effect on the organ weight during the 90-day oral toxicity test in (**A**) male and (**B**) female rats. Via oral gavage, animals were administered lyophilized probiotic powder at dosages of 250 (Low, L), 500 (Medium, M), and 1500 (High, H) mg/kg BW. Relative organ weights are presented as net organ weights normalized with body weights. Sterile water was used as the negative control. Data are presented as mean ± SD; *n* = 10.

**Figure 5 nutrients-16-00126-f005:**
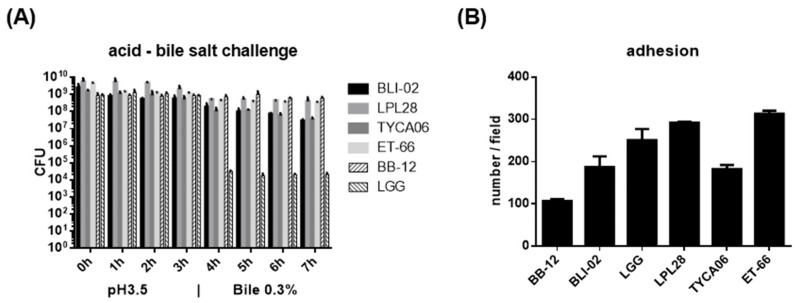
*B. longum* subsp. *infantis* BLI-02, *L. plantarum* LPL28, *L. acidophilus* TYCA06, and *L. paracasei* ET-66 displayed good probiotic potential. (**A**) BLI-02, LPL28, TYCA06, ET-66, and BB-12 well survived in 7 h continuous acidic and bile salt challenge. The colony forming unit (CFU) of each strain was recorded hourly in a pH 3.5 acidified environment for the first 3 h and in 0.3% bile salt for the next 4 h. Data were presented as mean ± SD of triplicate tests. (**B**) The intestinal adhesion of lactic acid bacteria was analyzed in human intestinal Caco-2 cells. More than 100 bacteria of BLI-02, LPL28, TYCA06, ET-66, and LGG were observed per field. Each group of bacteria was counted and averaged in seven view fields under a microscope with magnification × 1000. Data were presented as mean ± SD.

**Table 1 nutrients-16-00126-t001:** Antibiotic susceptibility profiles reported in MIC mg/mL using the microdilution method in accordance with ISO 10932 [47] guidelines ^a^.

Antibiotics	BLI-02	LPL28	TYCA06	ET-66
Gentamicin	8/64	4/16	4/16	2/32
Kanamycin	n.r.^b^	64/64	64/64	128/64
Streptomycin	16/128	16/n.r.	4/16	16/64
Tetracycline	0.5/8	16/32	0.5/4	2/4
Erythromycin	0.125/1	0.25/1	0.125/1	1/1
Clindamycin	0.063/1	0.063/2	0.5/1	0.25/1
Chloramphenicol	2/4	8/8	2/4	8/4
Ampicillin	0.125/2	0.25/2	1/1	2/4
Vancomycin	2/2	n.r. ^b^	1/2	n.r. ^b^

^a^ The result was presented as a measured value/cut-off value in ref. [46]. ^b^ n.r.: not required.

**Table 2 nutrients-16-00126-t002:** Acute oral toxicity test in Institute of Cancer Research (ICR) mice.

Strain	Gender	Dosage (g/kg BW)	Animal Number (*n*)	Week 0 (g)	Week 1 (g)	Week 2 (g)	Death Number (*n*)	LD_50_ (g/kg BW)
BLI-02	Male	10	10	19.17 ± 0.82	29.86 ± 1.26	35.04 ± 1.45	0	>10
	Female	10	10	17.04 ± 0.59	20.52 ± 1.06	21.86 ± 1.17	0	>10
LPL28	Male	10	10	18.83 ± 0.68	29.88 ± 1.18	35.09 ± 1.17	0	>10
	Female	10	10	17.15 ± 0.43	20.46 ± 0.73	21.58 ± 0.77	0	>10
TYCA06	Male	10	10	18.75 ± 0.62	29.27 ± 1.31	34.28 ± 1.72	0	>10
	Female	10	10	17.22 ± 0.57	20.67 ± 0.71	22.24 ± 0.82	0	>10
ET-66	Male	10	10	19.23 ± 0.62	30.08 ± 0.80	35.60 ± 0.95	0	>10
	Female	10	10	17.26 ± 0.64	20.84 ± 1.24	22.03 ± 1.29	0	>10

Abbreviations: BW, body weight; LD_50_, median lethal dose.

**Table 3 nutrients-16-00126-t003:** Mouse spermatocyte chromosomal aberration test.

Strain	Dosage (g/kg BW)	Animal Number (*n*)	Cell Number (*n*)	Chromosome Aberration (*n*)	Aberration Percentage (%)
Negative control	0 ^a^	5	500	6	0.8 ± 1.3
Positive control	0.04 ^b^	5	500	51	7.2 ± 1.3 **
BLI-02 (L)	2.5	5	500	9	1.8 ± 1.3
BLI-02 (M)	5	5	500	12	1.8 ± 1.3
BLI-02 (H)	10	5	500	13	1.6 ± 0.9
LPL28 (L)	2.5	5	500	7	1.0 ± 0.7
LPL28 (M)	5	5	500	14	1.8 ± 1.5
LPL28 (H)	10	5	500	11	1.6 ± 1.1
TYCA06 (L)	2.5	5	500	10	1.2 ± 1.6
TYCA06 (M)	5	5	500	13	2.0 ± 2.1
TYCA06 (H)	10	5	500	13	1.8 ± 1.3
ET-66 (L)	2.5	5	500	8	0.6 ± 1.3
ET-66 (M)	5	5	500	7	0.6 ± 0.5
ET-66 (H)	10	5	500	12	2.0 ± 1.4

Abbreviation: BW, body weight. ^a^ Sterile water as the negative control. ^b^ Cyclophosphamide (0.04 g/kg BW) as the positive control. ** Significant difference in comparison with the negative control group (** *p* < 0.01).

**Table 4 nutrients-16-00126-t004:** The hematological parameters in male and female rats after the 90-day oral gavage treatment at the highest dosage of 1.5 g/kg BW ^a^.

Parameter	Control	BLI-02	LPL28	TYCA06	ET-66
Male					
HB (g/L)	153 ± 6	138 ± 8	151 ± 5	153 ± 6	149 ± 5
RBC (×10^12^/L)	7.89 ± 0.40	6.85 ± 0.43	7.49 ± 0.40	7.95 ± 0.26	7.30 ± 0.39
HCT (%)	40.1 ± 1.9	38.6 ± 2.1	38.4 ± 1.6	39.6 ± 1.4	40.0 ± 1.4
WBC (×10^9^/L)	10.01 ± 0.98	9.58 ± 1.12	9.19 ± 0.93	9.43 ± 1.42	9.43 ± 1.07
PLT (×10^9^/L)	1056 ± 87	946 ± 99	1012 ± 69	1037 ± 73	978 ± 74
LYMPH (%)	82.7 ± 5.2	79.7 ± 7.6	73.6 ± 5.5	73.1 ± 6.1	74.1 ± 8.9
Neutrophil (%)	14.3 ± 0.7	14.6 ± 1.0	15.5 ± 0.7	15.0 ± 1.0	14.8 ± 0.8
Acidophil (%)	2.2 ± 0.4	2.1 ± 0.4	2.2 ± 0.4	2.1 ± 0.5	1.8 ± 0.4
Basophil (%)	0.4 ± 0.1	0.3 ± 0.1	0.4 ± 0.1	0.3 ± 0.1	0.3 ± 0.1
APTT (s)	32.0 ± 7.1	31.1 ± 3.4	33.0 ± 5.2	31.5 ± 4.1	32.3 ± 5.1
PT (s)	16.7 ± 2.2	16.1 ± 1.3	16.5 ± 1.5	17.0 ± 1.4	16.3 ± 1.2
Female					
HB (g/L)	158 ± 5	152 ± 7	151 ± 3	157 ± 4	154 ± 4
RBC (×10^12^/L)	7.75 ± 0.41	7.77 ± 0.36	7.42 ± 0.50	7.62 ± 0.46	7.86 ± 0.39
HCT (%)	39.4 ± 1.3	39.0 ± 1.5	40.1 ± 1.4	40.4 ± 1.3	39.0 ± 1.5
WBC (×10^9^/L)	8.59 ± 0.98	7.90 ± 1.40	7.73 ± 1.02	7.99 ± 1.43	7.68 ± 1.22
PLT (×10^9^/L)	1071 ± 86	1066 ± 79	1051 ± 68	1038 ± 78	1090 ± 84
LYMPH (%)	80.1 ± 6.6	77.0 ± 7.0	75.7 ± 5.4	77.6 ± 5.5	74.8 ± 6.6
Neutrophil (%)	14.1 ± 1.1	14.2 ± 0.7	13.7 ± 0.9	13.3 ± 0.6	14.0 ± 1.0
Acidophil (%)	1.4 ± 0.3	1.6 ± 0.5	2.0 ± 0.2	1.4 ± 0.3	1.5 ± 0.4
Basophil (%)	0.4 ± 0.1	0.4 ± 0.2	0.5 ± 0.1	0.4 ± 0.1	0.3 ± 0.1
APTT (s)	32.6 ± 5.0	32.1 ± 4.6	30.0 ± 4.5	31.1 ± 4.0	32.4 ± 4.3
PT (s)	17.1 ± 1.5	16.8 ± 1.4	16.4 ± 1.3	17.0 ± 1.4	16.2 ± 1.2

Abbreviations: HB, hemoglobin; RBC, red blood cells; HCT, hematocrit; WBC, white blood cells; PLT, platelet count; LYMPH, lymphocytes; APTT, activated partial thromboplastin time; PT, prothrombin time. ^a^ Mean ± SD; *n* = 10.

**Table 5 nutrients-16-00126-t005:** Serum biochemistry parameters in male and female rats after the 90-day oral gavage treatment at the highest dosage of 1.5 g/kg BW ^a^.

Parameter	Control	BLI-02	LPL28	TYCA06	ET-66
Male					
ALT (U/L)	48.6 ± 5.9	47.3 ± 7.3	42.4 ± 6.6	46.3 ± 7.2	46.0 ± 7.7
AST (U/L)	138.8 ± 16.3	142.4 ± 12.1	137.5 ± 7.1	138.5 ± 15.2	138.7 ± 10.9
ALP (U/L)	194.1 ± 9.9	195.5 ± 10.6	191.7 ± 12.3	198.7 ± 13.3	196.6 ± 10.1
γ-GT (U/L)	1.2 ± 0.4	1.3 ± 0.3	1.4 ± 0.4	1.3 ± 0.3	1.4 ± 0.3
Urea (mmol/L)	6.55 ± 0.91	5.93 ± 0.89	5.75 ± 0.48	6.42 ± 0.95	6.42 ± 0.93
CRE (μmol/L)	50.1 ± 4.5	49.5 ± 5.8	50.9 ± 4.3	49.8 ± 3.8	51.2 ± 4.8
GLU (mmol/L)	6.69 ± 0.67	6.99 ± 0.59	6.62 ± 0.55	6.58 ± 0.41	6.67 ± 0.43
TP (g/L)	50.0 ± 10.5	47.7 ± 10.0	51.2 ± 7.6	47.0 ± 11.1	49.6 ± 10.6
ALB (g/L)	22.7 ± 3.8	24.2 ± 3.0	21.0 ± 3.8	22.6 ± 4.0	21.4 ± 4.1
TC (mmol/L)	1.28 ± 0.42	1.53 ± 0.31	1.29 ± 0.28	1.31 ± 0.37	1.31 ± 0.32
TG (mmol/L)	0.95 ± 0.14	1.04 ± 0.20	0.99 ± 0.20	1.04 ± 0.23	1.10 ± 0.18
Cl (mmol/L)	125.1 ± 16.1	128.9 ± 16.8	121.9 ± 14.1	124.7 ± 11.8	124.8 ± 13.6
K (mmol/L)	5.43 ± 0.67	5.78 ± 1.08	5.85 ± 0.62	5.97 ± 1.00	5.21 ± 0.86
Na (mmol/L)	162.7 ± 16.1	162.4 ± 17.5	165.5 ± 14.9	160.9 ± 14.1	155.5 ± 15.9
Female					
ALT (U/L)	26.7 ± 3.6	31.8 ± 4.0	27.7 ± 4.5	27.7 ± 4.5	29.5 ± 5.7
AST (U/L)	111.0 ± 16.0	110.2 ± 15.2	103.4 ± 10.4	108.0 ± 16.2	108.9 ± 20.3
ALP (U/L)	140.9 ± 25.8	136.1 ± 25.9	149.3 ± 26.6	150.0 ± 26.3	148.1 ± 18.9
γ-GT (U/L)	1.4 ± 0.4	1.4 ± 0.3	1.2 ± 0.2	1.3 ± 0.3	1.4 ± 0.3
Urea (mmol/L)	7.63 ± 0.86	7.98 ± 0.81	7.51 ± 1.08	8.02 ± 1.10	7.43 ± 1.11
CRE (μmol/L)	52.1 ± 4.3	48.4 ± 3.6	48.7 ± 4.1	47.7 ± 4.4	51.1 ± 2.5
GLU (mmol/L)	6.73 ± 0.57	6.59 ± 0.54	6.76 ± 0.51	6.83 ± 0.68	6.97 ± 0.57
TP (g/L)	64.0 ± 16.1	67.7 ± 11.1	64.0 ± 11.9	66.1 ± 16.1	61.5 ± 14.6
ALB (g/L)	32.0 ± 5.0	30.4 ± 6.0	30.1 ± 4.9	27.3 ± 4.7	29.8 ± 4.0
TC (mmol/L)	1.47 ± 0.45	1.75 ± 0.44	1.79 ± 0.44	1.48 ± 0.42	1.46 ± 0.45
TG (mmol/L)	0.95 ± 0.24	1.06 ± 0.12	0.98 ± 0.23	1.00 ± 0.26	1.03 ± 0.21
Cl (mmol/L)	115.6 ± 9.0	113.3 ± 5.6	117.0 ± 5.0	114.9 ± 8.7	112.0 ± 6.8
K (mmol/L)	5.20 ± 0.72	4.72 ± 1.06	5.60 ± 1.15	5.50 ± 0.90	5.05 ± 0.90
Na (mmol/L)	151.7 ± 5.4	146.8 ± 5.1	152.0 ± 4.8	149.6 ± 7.9	146.1 ± 7.0

Abbreviations: ALT, alanine aminotransferase; AST, aspartate aminotransferase; ALP, alkaline phosphatase; γ-GT, gamma glutamyl transpeptidase; CRE, creatinine; GLU, glucose; TP, total protein; ALB, albumin; TC, total cholesterol; TG, triacylglycerol; Cl, chloride; K, potassium; Na, sodium. ^a^ Mean ± SD; *n* = 10.

**Table 6 nutrients-16-00126-t006:** Bacteriostatic activity tests of *B. longum* subsp. *infantis* BLI-02, *L. plantarum* LPL28, *L. acidophilus* TYCA06, and *L. paracasei* ET-66.

	BLI-02	LPL28	TYCA06	ET-66
*Vibrio parahaemolyticus* ^a^	>5 cm	>3 cm	>2 cm	>4 cm
*Helicobacter pylori* ^b^	16.37% ±3.26%	11.33% ±2.19%	64.77% ±20.70%	22.28% ±6.22%
*Aggregatibacter actinomycetemcomitans* ^a^	>2 cm	>2 cm	>2 cm	>3 cm

^a^ The modified agar overlay method. ^b^ The liquid culture assay.

## Data Availability

The datasets used and/or analyzed during the current study are available from the corresponding author upon reasonable requests.

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
