# Peer review of "Safety Assessment and Probiotic Potential Comparison of Bifidobacterium longum subsp. infantis BLI-02, Lactobacillus plantarum LPL28, Lactobacillus acidophilus TYCA06, and Lactobacillus paracasei ET-66"

_nutrients, 2023, doi:10.3390/nu16010126_

Round 1
Reviewer 1 Report
Comments and Suggestions for Authors
Dear Authors,
I have some corrections:
Abstract must be shortened (up to a maximum of 200 words)
- the type of graph for figure 1 is not chosen appropriately because it is difficult to read, use another format if possible
Author Response
Dear reviewer:
We would like to express our sincere gratitude for providing valuable feedback on our manuscript titled "Safety Assessment and Probiotic Potential Comparison of Bifidobacterium longum subsp. infantis BLI-02, Lactobacillus plantarum LPL28, Lactobacillus acidophilus TYCA06, and Lactobacillus paracasei ET-66."
We genuinely appreciate the time and attention you devoted to reviewing our manuscript. Your comments and suggestions have been thoroughly considered, and we are pleased to provide a detailed explanation of the modifications we have implemented based on your feedback. Followings are the point-to-point response to your comments.
Sincerely,
Hsieh-Hsun Ho
Comments 1: Abstract must be shortened (up to a maximum of 200 words)
Response 1: Thank you for reminding us about the length of the abstract. We have revised it to ensure conciseness and coherence.
Comments 2: The type of graph for figure 1 is not chosen appropriately because it is difficult to read, use another format if possible
Response 2: Thank you for pointing out the issues with the presentation of Figure 1. We have revised Figure 1 by incorporating color and adjusting the axis scales. We believe these adjustments will enhance readability.
Reviewer 2 Report
Comments and Suggestions for Authors
In this study, the authors assessed the safety and probiotic potential of four lactic acid bacteria strains via in vitro and in vivo experiments. While it is not clear whether the information reported here is new to the readership. Please revise the introduction to include more background on the four specific strains with a focus on the current understanding of their safety and probiotic potential.
Besides, the claim that "the lyophilize powder of these fours strains is safe as probiotic supplement" is too strong. The in vitro and mice experiments in this study could not provide sufficient evidence to support this conclusion.
Also, it is hard to identify specific groups in some of the figures due to the colors and patterns used, especially in Figure 4B.
Comments on the Quality of English LanguageThe language is generally in good quality.
Author Response
Dear reviewer:
We would like to express our sincere gratitude for providing valuable feedback on our manuscript titled "Safety Assessment and Probiotic Potential Comparison of Bifidobacterium longum subsp. infantis BLI-02, Lactobacillus plantarum LPL28, Lactobacillus acidophilus TYCA06, and Lactobacillus paracasei ET-66."
We genuinely appreciate the time and attention you devoted to reviewing our manuscript. Your comments and suggestions have been thoroughly considered, and we are pleased to provide a detailed explanation of the modifications we have implemented based on your feedback. Followings are the point-to-point response to your comments.
Sincerely,
Hsieh-Hsun Ho
Comments 1: In this study, the authors assessed the safety and probiotic potential of four lactic acid bacteria strains via in vitro and in vivo experiments. While it is not clear whether the information reported here is new to the readership. Please revise the introduction to include more background on the four specific strains with a focus on the current understanding of their safety and probiotic potential.
Response 1: Thank you for your valuable suggestions regarding the introduction. They have greatly contributed to improving the quality of our manuscript. We have provided additional information on the reasons for conducting safety studies on probiotics and enhanced the historical background of the four species of lactic acid bacteria, including the years of isolation for BLI-02, LPL28, TYCA06, and ET-66. Despite the classification of lactic acid bacteria as Generally Recognized as Safe (GRAS) for human consumption, the report published by the Food and Agriculture Organization (FAO)/World Health Organization (WHO) in 2002 emphasized the necessity of a systematic assessment of the safety of lactic acid bacteria. As BLI-02, LPL28, TYCA06, and ET-66 have demonstrated various functional properties, this study follows international guidelines for pharmaceutical and food safety assessments. High-dose toxicity experiments with these strains are conducted using animal models to investigate their safety. For details of the revised version, please see the attachment of the tracked revised manuscript.
Comments 2: Besides, the claim that "the lyophilize powder of these fours strains is safe as probiotic supplement" is too strong. The in vitro and mice experiments in this study could not provide sufficient evidence to support this conclusion.
Response 2: Thank you for your precious comments. The conclusion of the abstract has been revised to state, "the lyophilized powder of these four strains appears to be a safe probiotic supplement at tested dosages." The revised version could be found on line 28.
Comments 3: Also, it is hard to identify specific groups in some of the figures due to the colors and patterns used, especially in Figure 4B.
Response 3: Thank you for pointing out the issues with the presentation of Figure. We have revised Figure 1, Figure 3, and Figure 4 by incorporating color and adjusting the axis scales. We believe these adjustments will enhance readability.
Comments on the Quality of English Language: The language is generally in good quality.
Response: Thank you.
Round 2
Reviewer 2 Report
Comments and Suggestions for Authors
The comments and suggested changes have been properly addressed. Thank you for your efforts.